# A surface ocean pCO<sub>2</sub> product with improved representation of interannual variability using a vision transformer-based model

Xueying Zhang<sup>1</sup>, Enhui Liao<sup>1\*</sup>, Wenfang Lu<sup>2,3</sup>, Zelun Wu<sup>4</sup>, Guansuo Wang<sup>5,6</sup>, Xueming Zhu<sup>3</sup>, Shiyu Liang<sup>7\*</sup>

<sup>1</sup>School of Oceanography, Shanghai Jiao Tong University, Shanghai, 200240, China
 <sup>2</sup>School of Marine Sciences, State Key Laboratory of Environmental Adaptability for Industrial Products, Sun Yat-sen University, Zhuhai, Guangdong, 519082, China
 <sup>3</sup>Southern Marine Science and Engineering Guangdong Laboratory (Zhuhai), Zhuhai, Guangdong, 519082, China
 <sup>4</sup>School of Marine Science and Policy, University of Delaware, Newark, Delaware, 19716, USA
 <sup>5</sup>Observation and Research Station of Huaniaoshan East China Sea Ocean-Atmosphere Integrated Ecosystem, Ministry of

<sup>5</sup>Observation and Research Station of Huaniaoshan East China Sea Ocean-Atmosphere Integrated Ecosystem, Ministry of Natural Resources, Shanghai, 200137, China <sup>6</sup>East China Sea Forecasting and Hazard Mitigation Center, Ministry of Natural Resources, Shanghai, 200137, China

<sup>7</sup>John Hopcroft Center for Computer Science, Shanghai Jiao Tong University, Shanghai, 200240, China

Correspondence to: Enhui Liao (ehliao@situ.edu.cn) and Shiyu Liang (lsy18602808513@situ.edu.cn)

- Abstract. The ocean plays a crucial role in regulating the global carbon cycle and mitigating climate change, with the spatial distribution and temporal variations of ocean surface partial pressure of CO<sub>2</sub> (spCO<sub>2</sub>) directly determining the air-sea CO<sub>2</sub> flux. However, constructing a global spCO<sub>2</sub> data product that is able to resolve interannual and decadal variability remains a challenge due to the spatial sparsity and temporal discontinuity of observational data. This study presents an approach based on the Vision Transformer (ViT) model, combining high-quality observational data from the CO<sub>2</sub> Atlas (SOCAT) with
- multiple advanced global ocean biogeochemical models results to reconstruct a global monthly spCO<sub>2</sub> dataset (SJTU-AViT) at 1° resolution from 1982 to 2023. The approach employs the self-attention mechanism of the ViT model to enhance the modeling of the spatial and temporal variations of spCO<sub>2</sub>, as well as incorporates physical-biogeochemical constraints from the derivative of spCO<sub>2</sub> with respect to key controlling factors as additional features. The incorporation of advanced ocean biogeochemical models during the training process allows the ViT-based model to capture more accurate spCO<sub>2</sub> variability in
- these data-sparse regions. Evaluations demonstrate that the new data product effectively captures spCO<sub>2</sub> variability at both global and regional scales, showing good consistency with SOCAT observations, long-term ocean station data, and global atmospheric CO<sub>2</sub> trends. The reconstructed spCO<sub>2</sub> demonstrates strong capability in reproducing spCO<sub>2</sub> anomalies during El Niño-Southern Oscillation (ENSO) events, particularly in the eastern Pacific Ocean, where it shows a correlation of 0.81 with the Niño 3.4 index and demonstrates high consistency with cruise data. Based on the SJTU-AViT dataset, the estimated global
- air-sea CO<sub>2</sub> flux patterns are consistent with known regional features such as strong uptake in the Southern Ocean and 30 outgassing in the tropical Pacific. This study not only provide a new 42-year data product for advancing understanding of the ocean carbon cycle and global carbon budget assessments, but also introduces a new Transformer-based deep learning framework for Earth system data reconstruction. The data product is publicly accessible at https://doi.org/10.5281/zenodo.15331978 (Zhang et al., 2025) and will be updated regularly.

#### 35 1 Introduction

55

Global warming is primarily driven by the continuous increase in atmospheric greenhouse gas concentrations, with carbon dioxide (CO<sub>2</sub>) being the dominant contributor (Friedlingstein et al., 2023). The ocean, as one of the largest carbon sinks in the Earth system, absorbs approximately 25% of anthropogenic CO<sub>2</sub> emissions (~2.8 PgC yr<sup>-1</sup>), playing a crucial role in regulating the global carbon cycle and climate change (Friedlingstein et al., 2023). However, the ocean's capacity to absorb CO<sub>2</sub> is not constant; rather, it is influenced by a complex interplay of atmospheric CO<sub>2</sub> concentration, ocean physical and biogeochemical

- constant; rather, it is influenced by a complex interplay of atmospheric CO<sub>2</sub> concentration, ocean physical and biogeochemical processes, exhibiting significant spatiotemporal variability (Landschützer et al., 2016; Takahashi et al., 2002). Accurate estimation of oceanic CO<sub>2</sub> fluxes is therefore essential for understanding carbon cycle mechanisms and assessing the effectiveness of the ocean as a carbon sink.
- Accurately quantifying air-sea CO<sub>2</sub> flux relies on precise estimates of sea surface CO<sub>2</sub> partial pressure (spCO<sub>2</sub>). While the surface ocean CO<sub>2</sub> atlas (SOCAT) database (Bakker et al., 2016) provides a valuable foundation, observational coverage remains sparse and uneven, particularly in high-latitude regions and during winter months when harsh sea conditions limit measurements (Mackay and Watson, 2021). Existing approaches for estimating spCO<sub>2</sub> primarily fall into two categories: numerical biogeochemical modeling and data-driven methods. Traditional numerical biogeochemical models simulate spCO<sub>2</sub> by parameterizing physical and biogeochemical processes (Kern et al., 2024; Roobaert et al., 2022). However, due to the highly
- nonlinear dynamics of the oceanic carbon cycle and regional heterogeneity, numerical biogeochemical models still exhibit considerable uncertainties in reconstructing the spatiotemporal distribution of spCO<sub>2</sub> (Rödenbeck et al., 2015; Roobaert et al., 2022). Moreover, simplified parameterization of biogeochemical processes may lead to underestimation or overestimation of oceanic carbon uptake, ultimately affecting the accuracy of global carbon budget assessments (Resplandy et al., 2024).

To address these limitations, statistical interpolation and machine learning techniques have been increasingly employed to reconstruct spCO<sub>2</sub> distributions based on available observations (Rödenbeck et al., 2015). Statistical interpolation methods,

- such as regression-based approaches (Rödenbeck et al., 2015), Bayesian techniques (Valsala et al., 2021), and tree-based algorithms (Geurts et al., 2006), leverage the spatiotemporal correlation of spCO<sub>2</sub> observations and have achieved moderate success in some regions (Gregor et al., 2019). However, these methods struggle with poor reconstruction accuracy in data-sparse regions and do not fully capture the complex ocean carbon biogeochemical processes effectively (Hauck et al., 2023).
- Consequently, machine learning approaches have gained prominence in recent years. In particular, feedforward neural networks (FFNNs) have demonstrated superior reconstruction accuracy and have become one of the most widely used tools for spCO<sub>2</sub> and other ocean data estimation (Denvil-Sommer et al., 2019; Landschützer et al., 2013; Zeng et al., 2014). These methods yield root mean square errors (RMSE) of approximately 18 μatm in open ocean regions, aligning well with SOCAT observations (Gregor et al., 2019).
- Despite recent advances, significant challenges remain in reconstructing spCO<sub>2</sub>, particularly in capturing its interannual and decadal variability, which plays a pivotal role in modulating oceanic carbon uptake. Accurate characterization of this variability remains a central issue in the ocean carbon field. Furthermore, the widely used FFNNs method may introduce discontinuities

at cluster boundaries due to the discrete nature of data grouping, impacting the representation of spCO<sub>2</sub> variability (Gregor et al., 2019). These discontinuities often require additional post-smoothing procedures, which may introduce artificial bias,
thereby increasing reconstructed data uncertainty or suppressing real spatiotemporal variability (Gregor et al., 2019). More broadly, a persistent imbalance of approximately 1 Pg C yr<sup>-1</sup> remains in the global carbon budget, reflecting unresolved discrepancies between estimated sources and sinks on the global scale. One plausible contributor to this imbalance is the inadequate characterization of the interannual variability in oceanic carbon uptake (Friedlingstein et al., 2023). Therefore, this study develops a novel reconstruction method to more accurately capture interannual dynamics, alleviate artificial spatial
discontinuities, particularly across cluster boundaries, and ultimately contribute to close the global carbon budget (Rödenbeck

et al., 2015).

Transformer architectures, originally developed for sequence modeling in natural language processing, have demonstrated exceptional capabilities in capturing long-range dependencies and learning complex, nonlinear relationships across highdimensional datasets. Their scalability and effectiveness in tasks such as machine translation, language understanding, and

80 large language models (e.g., Chat-GPT) have established them as a cornerstone of modern artificial intelligence. Recently, these models have been extended to atmospheric science and oceanography, where they have shown promising performance in forecasting ocean states and extracting spatiotemporal patterns from large-scale environmental data. Given these advantages, Transformer-based frameworks offer considerable potential for data reconstruction in oceanography, where challenges such as sparse observations, multiscale variability, and strong spatiotemporal coupling demand flexible and powerful modeling

```
approaches (Ji et al., 2025; Liu et al., 2024).
```

Against this backdrop, the image-based Vision Transformer (ViT) architecture, with its multi-head self-attention mechanism and high representational capacity, has emerged as a powerful tool for capturing the complex spatiotemporal features of oceanic environmental variables. This model is well-suited for reconstructing spCO<sub>2</sub>, as it can integrate diverse environmental drivers such as sea surface temperature (SST), salinity (SSS), chlorophyll concentration (Chl-a), mixed layer depth (MLD),

- and atmospheric CO<sub>2</sub> concentration. To enhance the physical constraints of spCO<sub>2</sub> reconstruction, this study incorporates ocean carbonate system sensitivities to key variables like SST, SSS, dissolved inorganic carbon (DIC), and total alkalinity (ALK) (Takahashi et al., 1993). In this context, multi-stage training strategies that combine simulated data from Earth system models and observational constraints have also proven effective in improving model robustness and accuracy. The spCO<sub>2</sub>-based Shanghai Jiao Tong University aggregation Vision Transformer (SJTU-AViT) developed in this study effectively captures
- both spatial variations and interannual to decadal variability of ocean carbon dynamics at global scales. This contributes to enhancing our understanding of the temporal dynamics of oceanic carbon uptake and addressing imbalances in the global carbon budget.

#### 2 Data and methods

#### 2.1 Training Data Description

This study selects a range of input features for model training to comprehensively capture the dynamics of surface ocean spCO<sub>2</sub> variability through sensitivity tests and other spCO<sub>2</sub> data reconstruction studies (Denvil-Sommer et al., 2019; Landschützer et al., 2013; Zeng et al., 2014). The selected input features include SST, SSS, Chl-a, MLD, and air CO<sub>2</sub>. Additionally, we introduce physical constraints based on the relationship

$$\Delta spCO_2 \approx \frac{\partial spCO_2}{\partial DIC} \Delta DIC + \frac{\partial spCO_2}{\partial ALK} \Delta ALK + \frac{\partial spCO_2}{\partial SST} \Delta SST + \frac{\partial spCO_2}{\partial SSS} \Delta SSS$$
(1)

- that the sensitivities of CO<sub>2</sub> partial pressure to SSS, SST, DIC, and ALK  $\left(\frac{\partial spCO_2}{\partial SSS}, \frac{\partial spCO_2}{\partial SST}, \frac{\partial spCO_2}{\partial DIC}, \frac{\partial spCO_2}{\partial ALK}\right)$  are included as input features in the deep learning model to reinforce spCO<sub>2</sub> physical-biogeochemical consistency (Takahashi et al., 1993). These parameters represent key physical, chemical, and biological factors influencing the distribution of spCO<sub>2</sub> in the ocean. All the input features are interpolated into a uniform 1°×1° spatial resolution and monthly temporal resolution.
- The input datasets consist of long-term time series and high-resolution spatial data, ensuring both temporal and spatial consistency across variables (Table 1). SST data were obtained from the NOAA Optimum Interpolation SST (OISST) (version v02r01) dataset, spanning from 1982 to 2023 with daily resolution and a spatial resolution of 0.25° (Reynolds et al., 2007; Huang et al., 2021). Sea surface salinity (SSS) data were sourced from the Hadley Centre EN.4.2.2 (c14) dataset, covering the period from 1982 to 2023 with daily resolution and a spatial resolution of 0.25° (Good et al., 2013). Chl-a data were derived from the European Space Agency Climate Change Initiative (ESA CCI) Ocean Colour (version 5.0) dataset, spanning 1997 to
- 2022 with daily resolution and a spatial resolution of 4 km (Jackson et al., 2017). Ocean MLD data were obtained from the World Ocean Circulation Experiment (WOCE) Global Data Version 3.0, providing monthly climatology with a spatial resolution of 2° (de Boyer Montégut et al., 2004). Atmospheric CO<sub>2</sub> mole fraction (xCO<sub>2</sub>) data were sourced from the NOAA Earth System Research Laboratories (ESRL) marine boundary layer (MBL) CO<sub>2</sub> product, covering the period from 1982 to 2023 with about 8-day resolution and meridional spacing (Dlugokencky et al., 2019).
- The monthly climatologies of  $\frac{\partial spCO_2}{\partial SSS}$ ,  $\frac{\partial spCO_2}{\partial SST}$ ,  $\frac{\partial spCO_2}{\partial DIC}$ ,  $\frac{\partial spCO_2}{\partial ALK}$  at a spatial resolution of 1° are included as additional input features, sourced from the ocean-driven global biogeochemical model simulations (Liao et al., 2020). These rate-of-change variables help to reflect the influences of temperature, salinity, alkalinity, and DIC on spCO<sub>2</sub>, thereby enriching the deep learning model's representation of the underlying biogeochemical processes. Additionally, spCO<sub>2</sub> from the SOCAT database was used as the target variable for the model training and validation. The SOCAT dataset used in this study is version 2024

The Coupled Model Intercomparison Project Phase 6 (CMIP6) model results are downloaded from the Lawrence Livermore National Laboratory node database (https://esgf31node.llnl.gov/projects/cmip6/, at the time of this study). We selected a subset of 7 ESMs based on the availability of download access through our cluster and the availability of environmental variables (see Supplement Section S2 for details). The biogeochemical model adopted in this study is from the Geophysical Fluid

(Fig. S1) which is interpolated into the uniform  $1^{\circ} \times 1^{\circ}$  spatial resolution and monthly temporal resolution (Bakker et al., 2016).

Dynamics Laboratory (GFDL). The model includes Modular Ocean Model version 6 (MOM6), sea ice simulator version 2, carbon ocean biogeochemistry, and lower trophics version 2 (COBALT v2), which is collectively referred to as MOM6-COBALT2 (Adcroft et al., 2019; Stock et al., 2020). The model performance is thoroughly assessed, and it reproduces well-observed physical and biogeochemical features in the global ocean (Stock et al., 2020). More detailed model evaluations and configurations, including spin-up, atmospheric forcing, and initial conditions, can be found in Liao et al. (2020).

| 1 | 1   | ~ |
|---|-----|---|
|   | - 4 | ~ |
| 1 | Э   | 2 |

| Variable                                           | Units              | Period    | Resolution              | Dataset                               | reference                          |
|----------------------------------------------------|--------------------|-----------|-------------------------|---------------------------------------|------------------------------------|
| Atmospheric CO <sub>2</sub><br>(xCO <sub>2</sub> ) | ppm                | 1982-2023 | Meridional,<br>monthly  | ESRL MBL CO <sub>2</sub> product      | Dlugokencky et al.<br>(2019)       |
| Chlorophyll a (Chl a)                              | mg m <sup>-3</sup> | 1997-2022 | 4 km, daily             | ESA CCI Ocean Colour<br>(Version 5.0) | Jackson et al. (2017)              |
| Sea surface<br>temperature (SST)                   | °C                 | 1982-2023 | 0.25°, daily            | NOAA OISST (Version<br>v02r01)        | Reynolds et al. (2007)             |
| Sea surface salinity<br>(SSS )                     | PSU                | 1982-2023 | 0.25°, daily            | Hadley Center EN.4.2.2<br>(c14)       | Good et al. (2013)                 |
| Ocean mixed layer<br>depth (MLD)                   | m                  | 12 month  | 2°, monthly climatology | WOCE Global Data<br>Version 3.0       | de Boyer Montegut et al.<br>(2004) |
| SOCAT                                              | µatm               | 1982-2023 | 1°, monthly             | SOCAT version 2024<br>data products   | Bakker et al. (2016)               |

#### **2.2 Model Architecture**

The deep learning model employed in this study is a Vision Transformer (ViT, Fig. 1), originally proposed by Dosovitskiy et al. (2020) for capturing spatial dependencies in large-scale image-like datasets. The design of ViT tackled the key limitation of the CNN-like methods, which implies the translation-invariant property of learned kernels. This property failed to learn the remote connections across regions among multiple variables (Liu et al., 2024). The ViT model employs a self-attention mechanism to capture long-term connections and complex spatial and temporal patterns (Nguyen et al., 2023), allowing it to dynamically adjust its receptive field and capture both localized details and large-scale variations. As a result, the model is able to provide a more comprehensive characterization of the relationships between spCO<sub>2</sub> and oceanic variables across spatial

## 145 scales.

The ViT-based framework for spCO<sub>2</sub> reconstruction includes four main steps. The first is variable tokenization, a process that involves partitioning the input data into local regions. Each region is treated as an image patch for subsequent processing and feature extraction (Dosovitskiy et al., 2020). These input variables are standardized and formatted into a multi-channel

input to ensure feature extraction occurs on a unified scale. Then, the ocean fields are segmented into fixed-size image patches.
For example, the SST field (180×360) is divided into non-overlapping 6×6 grids on every patch, resulting in 30×60 patches. The data in each patch is then projected into a high-dimensional vector through a patch embedding layer, preserving critical spatial structures and providing a suitable input representation for the Transformer framework.

The second step is variable aggregation, where a cross-attention mechanism is employed to integrate information across multiple environmental input variables (Vaswani et al., 2017). Given that different variables influence spCO<sub>2</sub> through distinct

- mechanisms, other methods like simple concatenation may obscure crucial dynamic relationships. The cross-attention mechanism enables the model to adaptively assign appropriate weights to different variables, emphasizing those that contribute most significantly to spCO<sub>2</sub> variations (Jaegle et al., 2021). To further enhance its ability to capture spatiotemporal dynamics, the model incorporates position encoding and time encoding at this stage, ensuring temporal consistency in the input data and improving the interpretability of ocean carbon cycle processes (Wu et al., 2021).
- The third step is Transformer backbone, where the data are fed into a Transformer backbone composed of 10 stacked Transformer blocks. Each block integrates multi-head self-attention (16 heads), layer normalization (LayerNorm), and a feedforward neural network (MLP) (Dosovitskiy et al., 2020; Vaswani et al., 2017). The multi-head self-attention mechanism enables the model to learn long-range dependencies and capture complex spatial interactions by attending to multiple representation subspaces simultaneously—an essential feature for modeling the inherently spatiotemporal dynamics of
- oceanographic variables. To further enhance representation learning, linear transformation and concatenation operations (Linear & Concat) are employed across layers. These operations support deep feature fusion, enabling the network to integrate both fine-scale local variations and broader climate-driven signals.

The final step is the model output. This step incorporates a pooling head for dimensionality reduction, producing the global oceanic spCO<sub>2</sub> fields as the output. The loss function is minimized by comparing the reconstructed values against observational

datasets, ensuring both physical consistency and numerical accuracy. The ViT-based model consists of approximately 115 million parameters and is trained in parallel across eight NVIDIA RTX 4090 GPUs, with each epoch requiring approximately 10 minutes.

To enhance model performance, we employ a multi-stage training strategy. First, we pre-train the ViT-based model using the 7 CMIP6 model results to learn a general relationship between spCO<sub>2</sub> and the environmental variables (SST, SSS, Chl-a,

- MLD, and air CO<sub>2</sub>). We then fine-tune the ViT-based model using data from the ocean-driven global ocean biogeochemical models (e.g., MOM6-COBALT) and further refine it with SOCAT observations to improve accuracy and applicability. The incorporation of the CMIP6 model and advanced ocean biogeochemical models enhances the spCO<sub>2</sub> reconstruction by mitigating the data sparsity issue, particularly in regions with limited observations, such as the Indian Ocean and high-latitude areas. Through the use of transfer learning, the model can better leverage global climate data to fill gaps in observational
- coverage.

Figure 1. Schematic of the Vision Transformer (ViT)-based framework for spCO2 reconstruction. The framework includes four main steps. The first is variable tokenization, where the input oceanographic variables (e.g., SST, SSS, Chl-a, MLD, and atmospheric 185 CO<sub>2</sub>) are divided into spatial patches and passed through a convolutional embedding layer. The second step is variable aggregation, where multiple variables are aggregated into one vector through the cross-attention mechanism. The third step is Transformer backbone, where the data are passed through stacked Transformer blocks that incorporate multi-head self-attention, layer normalization, and feedforward neural networks to capture complex spatiotemporal dependencies. The final step is model output, where a pooling head aggregates the learned representations and generates the spCO<sub>2</sub> fields.

#### 190 2.3 Validation Procedure and Data

following sections.

The SOCAT dataset was randomly divided into 80% for training and 20% for validation. For the independent test at long-term stations, data from these stations were excluded, and the model was trained using the remaining SOCAT data. In the final results generation phase, the full SOCAT dataset was utilized to produce the spCO<sub>2</sub> estimates. These estimates are subsequently used for analyses of climatological states, seasonal variations, and interannual changes in spCO<sub>2</sub>. In the comparison with SOCAT data, SJTU-AVIT values are first interpolated to match the spatial and temporal locations of SOCAT observations. Subsequently, the climatological mean, seasonal variations, and interannual changes are calculated at each grid point where data are available. The processed SJTU-AViT data are then compared with the corresponding SOCAT observations in the

200

195

In the training process, we adopt the latitude-weighted mean squared error (MSE) as the loss function to ensure that the model accommodates the spatial variability caused by the Earth's curvature. The latitude-weighted MSE effectively emphasizes the prediction accuracy in low-latitude regions, which occupy a larger proportion of the Earth's surface (Nguyen et al., 2023; Willard et al., 2024). The loss function is computed as follows:

$$MSE = \frac{1}{N} \frac{1}{H} \frac{1}{W} \sum_{t=1}^{N} \sum_{h=1}^{H} \sum_{w=1}^{W} \alpha(h) (y_{t,h,w} - y_{obs,t,h,w})^2$$
(2)

where N is the total number of time points in the dataset, H and W are the numbers of latitudinal and longitudinal grid points, 205 respectively, and t, h, and w represent the time, latitude, and longitude indices, respectively.  $y_{obs,t,h,w}$  is the observed value, and  $y_{t,h,w}$  is the predicted value. The term  $\alpha(h)$  is the latitude weight.

In the validation process, we use multiple evaluation metrics, including mean bias error (MBE), mean absolute error (MAE), root mean square error (RMSE), and coefficient of determination ( $R^2$ ). These metrics have been extensively used in reconstructed data assessments and climate model evaluations. It is computed as follows:

210 
$$MBE = \frac{1}{n} \sum_{i=1}^{n} (y_{rec,i} - y_{obs,i})$$
 (3)

$$MAE = 1/n \sum_{i=1}^{n} |y_{rec,i} - y_{obs,i}|$$
(4)

$$RMSE = \sqrt{1/n\sum_{i=1}^{n} (y_{rec,i} - y_{obs,i})^2}$$
(5)

$$R^{2} = 1 - \sum_{i=1}^{n} \left( y_{obs,i} - y_{rec,i} \right)^{2} / \sum_{i=1}^{n} \left( \left( y_{obs,i} - \overline{y_{obs}} \right)^{2} \right)^{2}$$
(6)

where *n* repr