# Peer review of "A surface ocean pCO2 product with improved representation of interannual variability using a vision transformer-based model"

_Earth System Science Data, 2025_

## Author Comment (AC1)

**Response to Referee #1**

**General comments:**

This manuscript introduces a novel machine learning framework (SJTU-AViT) for reconstructing global sea surface $pCO_2$ at 1°×1° monthly resolution over the period 1982–2023. By incorporating physical–biogeochemical constraints as derived features, the approach enhances the quality of ocean carbon data reconstruction. The evaluation is comprehensive, covering mean states, seasonal cycles, and interannual variability, and shows strong skill in reproducing ENSO-related signals. This study makes a substantial contribution by providing a valuable new ocean carbon data product for the ocean carbon community and a useful machine learning framework in the field of ocean data reconstruction. The subject is highly relevant to the scope of Earth System Science Data. However, I have several general and specific comments and suggestions that should be addressed before the manuscript can be considered for publication.

We sincerely thank the reviewer for the constructive and insightful comments, which have greatly improved the quality and clarity of our manuscript. The reviewer's main concerns focused on the independence of training and validation data and its impact on the robustness of our results. To address these concerns, we supplemented analyses using independent datasets, confirming the robustness of the assessed interannual variability. We also clarified and expanded methodological details, data processing, training settings, and figures/tables to enhance transparency and reproducibility. Specifically, we have made the following revisions:

- Clarify the train/test split strategy with added spatiotemporal distribution maps, and validate model generalization against independent long-term stations (major comment #1, #5, minor comments #7).
- Address concerns on model–validation dependence by recalculating detrended/deseasoned STD using MPI-SOM-FFN trends and seasonal cycles, confirming robust spatial patterns beyond data dependence (major comment #2).
- Enhance methodological clarity by adding a process flowchart, specifying SOCAT gridded comparison and interpolation procedures, and clarifying variable standardization (major comment #3, #6, minor comments #1).
- Clarify input coverage by filling pre-1997 and 2023 gaps in Chl-a with climatology, using climatological MLD as input data with noted limitations, detailing $xCO_2$ (MBL) mapping (major comment #7, #8, minor comments #5).
- Implement minor edits and clarifications to enhance precision and consistency throughout the text, as detailed in the point-by-point response (major comment #4, minor comments #1-17).

Please see the detailed response below.

**Major comments:**

1. The Methods section (Model training & testing) should more clearly describe how the data were split into training and testing sets, along with the sample size

distribution. This information is essential for evaluating the model's generalization ability. The authors should specify whether the split was random, temporal, or spatial (e.g., by cruise lines or fixed stations). They should also report the number or proportion of samples in each subset, ideally stratified by time (e.g., decades) and/or region. Such details would improve transparency and reproducibility.

We have clarified the data partitioning procedure in the section 2.3 and have included additional analyses to support transparency and robustness of the evaluation.

Specifically, SOCAT samples were randomly split into training and test subsets, with 80% (277,528 samples) allocated for model training and 20% (69,142 samples) reserved as an independent test set. All random operations were conducted using a fixed seed (seed = 42) to ensure full reproducibility. The detailed split procedure and exact sample counts are now explicitly documented in the revised text (new Fig. 2 and lines 208-209), now as *"The SOCAT dataset was randomly divided into 80% (277,528 samples) for training and 20% (69,142 samples) for validation, using a fixed random seed (seed = 42) to ensure reproducibility."*.

The test procedure is included in the new reconstruction workflow (Fig. 2) and described in section S5 of the supplement (see explanation in our response to comment 3), and Figure S1 has also been revised to clearly illustrate the temporal and spatial distributions of the training and test sets. In addition, we additionally evaluated its performance against nine independent long-term observation stations that are not included in the SOCAT dataset. These stations provide continuous time series and serve as an independent benchmark. The results indicate that the model reliably captures both temporal variability and long-term trends, providing strong evidence of its generalization capability beyond the original training data.

[Figure]

Figure R1. (Figure S1 in supplement section S4). Data availability for spCO$_2$ reconstruction. (a) Spatial distribution of the number of all spCO$_2$ data points. (b) Annual data count of all spCO$_2$ data points over the period from 1982 to 2023. (c) Spatial distribution of the number of spCO$_2$ data points used for training. (d) Annual data count used for training over the period from 1982 to 2023. (e) Spatial distribution of the number of spCO$_2$ data points used for validation. (f) Annual data count used for validation over the period from 1982 to 2023.

2. The manuscript fills gaps in SOCAT observations using long-term trends and seasonal cycles from SJTU-AViT, followed by residual analysis to assess interannual variability. This procedure raises concerns about the lack of independence between the model and the validation data, since part of the evaluation relies on model-derived estimates. The authors should clarify and quantify the impact of this approach. For instance, they could limit the analysis to grid points or stations with continuous records, or apply long-term trends and seasonal cycles from an independent product to compare robustness. Demonstrating consistent results across methods would enhance the credibility of the conclusions.

To address the concern regarding the potential lack of independence between the model and the validation data, we conducted an additional analysis using an

independent reconstructed data product from MPI-SOM-FFN (Landschützer et al. 2016). This validation results were provided in the supplement material (section S5.8). Specifically, when calculating the detrended and deseasonalized SOCAT STD, we applied the long-term trends and seasonal cycles derived from the MPI-SOM-FFN data product instead of the SJTU-AViT estimates. The results, shown in the Fig. R2, demonstrate that the overall spatial distribution of SOCAT STD remains highly consistent, with only minimal deviations (1.68 µatm). This indicates that the small deviations observed between SJTU-AViT and SOCAT are not artifacts of model-data dependence. Therefore, the analysis confirms the robustness of our methodology and supports the credibility of the interannual variability assessment.

.

[Figure]

Figure R2. (Figure S15 in supplement section S5). Comparison of spCO$_2$ standard deviations on timescales longer than one year between SJTU-AViT, SOCAT, and MPI-SOM-FFN data product. (a) Standard deviation of spCO$_2$ from the SJTU-AViT at SOCAT observation grid points. (b) Standard deviation of spCO$_2$ from SOCAT data (the long-term trends and seasonal cycles derived from the SJTU-AViT). (c) Standard deviation bias between SJTU-AViT and SOCAT (panel a minus panel b). (d) Standard deviation of spCO$_2$ from the SJTU-AViT at SOCAT observation grid points. (e) Standard deviation of spCO$_2$ from SOCAT data (the long-term trends and seasonal cycles derived from the MPI-SOM-FFN). (f) Standard deviation bias between SJTU-AViT and SOCAT (panel d minus panel e).

3. To help readers better understand the implementation of the spCO$_2$ data reconstruction, I recommend adding a schematic figure in the main text that illustrates the reconstruction process based on the ViT model. Such a figure would improve both the readability of the manuscript and the clarity of the methodology.
We have added a schematic framework figure (Fig. 2) in the section 2.2, along with detailed description in the main text (lines 186-188) and supplement (section S5.1, lines 128-140), to illustrate the spCO$_2$ reconstruction workflow based on the ViT framework.

The summary description is presented in the main text (lines 186-188) as *"The*

*overall workflow of this multi-stage training strategy is summarized in Fig. 2, which also provides a schematic overview of the spCO$_2$ reconstruction workflow based on the ViT framework. The figure clearly visualizes the main steps, from data preprocessing through model training to evaluation (see detailed description in section S5.1).".*

The detailed description is presented in the supplement (section S5, lines 128-140) as *"The spCO$_2$ reconstruction workflow based on the ViT framework is organized into four main stages—Data Processing, Model Architecture, Training & Validation, and Evaluation & Analysis—as illustrated in Fig. 2 (in main text). At the top, the data processing panel shows the input sources (CMIP6, MOM6, SOCAT) and the preprocessing steps: temporal harmonization to a monthly cadence, spatial regridding to a 1°×1° grid, and feature normalization. These boxes indicate that all inputs are brought to a common spatio-temporal grid and scale before being passed to the model. The model architecture panel depicts how physical variables are converted into model inputs: variable tokenization, variable aggregation, and then fed into a Transformer backbone that learns spatial and temporal dependencies. The model output block illustrates that the network predicts monthly spCO$_2$ on the same 1° grid. The training & validation panel summarizes our multi-stage training strategy: (i) pretraining on CMIP6-derived fields, (ii) fine-tuning using MOM6 plus 80% of SOCAT, and (iii) evaluation using a withheld 20% SOCAT validation split and independent tests at long-term station sites. Finally, the evaluation & analysis panel shows the main evaluation products derived from the reconstruction: model performance metrics, climatology, seasonal cycle, interannual variability, and downstream analyses (air-sea CO$_2$ flux calculation and uncertainty analysis).".*

[Figure]

Figure R3. (Figure 2 in main text). Workflow of the spCO$_2$ reconstruction using the ViT-based framework. The workflow consists of four major stages: (a) Data processing, where CMIP6, MOM6, and SOCAT inputs are temporally harmonized, spatially interpolated, and normalized; (b) Model architecture, where variables are tokenized, aggregated into spatio-temporal embeddings, and processed by a Transformer backbone to predict monthly spCO$_2$; (c) Training and validation, involving CMIP6 pretraining, MOM6 and SOCAT fine-tuning, and evaluation against withheld SOCAT data and long-term stations; and (d) Evaluation and analysis, where model performance metrics, climatology, seasonal cycles, and interannual variability are assessed, leading to downstream analyses such as air-sea CO$_2$ flux estimation and uncertainty analysis (see detailed description in supplement section S5.1).

4. It is recommended that the authors include skill distribution tables in the supplement, stratified by ocean basin and latitude band. These tables should report, for each group, the sample size (N), R², RMSE, MAE, and MBE. Such quantitative evidence would support the statement that "biases are larger at high latitudes" and clearly demonstrate regional and latitudinal variations in model performance.

We have added skill distribution tables in the supplement (Table S4-S5), stratified by ocean basin and latitude band. For each group, we report the sample size (N), R², RMSE, MAE, and MBE. These tables provide quantitative evidence supporting the

statement that biases are larger at high latitudes and clearly illustrate the regional and latitudinal variations in model performance, as followed.

Based on the statistics, the skill distribution reveals distinct regional and latitudinal differences. Among ocean basins, the Pacific shows the best performance (N=159,783; $R^2$=0.94), while the Indian Ocean, despite its smaller sample size (N=6,354), also exhibits strong skill ($R^2$=0.95; RMSE=5.31). In contrast, the Atlantic performs relatively weaker with a lower correlation and a slight negative bias (N=111,326; $R^2$=0.81). The Arctic (N=10,316; RMSE=8.80) and Southern Ocean (N=48,636; RMSE=8.20) show notably larger errors and systematic negative biases, indicating a tendency of underestimation in polar regions. When stratified by latitude bands, errors are markedly larger at high latitudes, particularly in 60°S-90°S (N=16,602; $R^2$=0.86) and 60°N-90°N (N=30,802; $R^2$=0.92). By comparison, the tropics and subtropics exhibit smaller errors, such as 0–30°S (N=35,804; $R^2$=0.97). The 0-30°N band shows moderate error levels (RMSE=6.13) but a lower correlation ($R^2$=0.72), likely reflecting observational variance and sample characteristics. Overall, these quantitative results directly support our conclusion that biases are more pronounced at high latitudes. As discussed in the main text, this pattern can be attributed to the complexity of seasonal amplitudes and boundary processes (e.g., sea-ice cover and mixed layer variability), the limited representativeness and accuracy of forcing fields and input data in polar regions, and uneven observational coverage, all of which can amplify errors and biases.

Table R1 (Table S4 in supplement section S3). Skill metrics of the reconstructed spCO$_2$ by ocean basin.

| Ocean basin | N | $R^2$ | RMSE | MAE | MBE |
|---|---|---|---|---|---|
| Pacific ocean | 159783 | 0.94 | 6.79 | 5.29 | 0.30 |
| Atlantic ocean | 111326 | 0.81 | 7.10 | 5.31 | -0.31 |
| Indian ocean | 6354 | 0.95 | 5.31 | 4.75 | -0.08 |
| Arctic ocean | 10316 | 0.90 | 8.80 | 7.58 | -0.24 |
| Southern ocean | 48636 | 0.88 | 8.20 | 6.76 | -0.55 |

Table R2 (Table S5 in supplement section S3). Skill metrics of the reconstructed spCO$_2$ by latitude band.

| latitude band | N | $R^2$ | RMSE | MAE | MBE |
|---|---|---|---|---|---|
| 60°N-90°N | 30802 | 0.92 | 9.23 | 7.58 | -0.56 |
| 30°N-60°N | 123357 | 0.91 | 9.13 | 6.40 | 0.07 |
| 0-30°N | 96608 | 0.72 | 6.13 | 4.74 | 0.04 |
| 0-30°S | 35804 | 0.97 | 5.70 | 4.96 | -0.07 |
| 30°S-60°S | 43497 | 0.90 | 6.13 | 5.29 | -0.20 |
| 60°S-90°S | 16602 | 0.86 | 11.80 | 9.29 | -1.03 |

5. Regarding the independent test sites, it is recommended to provide a clear description in the main text along with detailed information. In the appendix, the nine observation stations used for independent testing should be listed, including their

names, geographic locations, observation periods, and the number of samples at each site. Since the BAT site does not have direct pCO₂ observations, please clarify the method used to calculate its monthly mean $pCO_2$ and specify the data sources for all sites.

(1) In the revised manuscript, we have added an appendix table and provided a clearer description of the independent test sites (section 2.3 (lines 215-220), and Table S3). We explicitly detail the nine long-term observation stations used for independent testing, including their names, geographic locations, observation periods, number of samples, and data sources. To facilitate visual interpretation, we have also included a map (supplement Fig. S2) showing the locations of the stations.

Lines 215-220 now reads as *"For the independent test at long-term stations, reconstructed values were extracted at the corresponding station locations using bilinear spatial interpolation, which incorporates information from surrounding grid cells to provide smoother and more representative estimates, and skill metrics were subsequently computed to evaluate model performance. Detailed information for these stations, including their names, geographic locations, observation periods, number of samples, and data sources, is provided in supplement Table S3, and their locations are shown in supplement Fig. S2 to facilitate visual interpretation."*.

(2) For the BAT station, which does not have direct $pCO_2$ observations, the monthly mean $pCO_2$ was calculated using the Python version of CO2SYS (PyCO2SYS; Humphreys et al., 2022). In the calculation, we used the carbonate dissociation constants from Waters and Millero (2013) (k1k2 = 15), the $KSO_4$ constant from Dickson (1990) (kso4 = 1), the HF dissociation constant from Perez and Fraga (1987) (hf = 2), and the total pH scale (pHscale = 1). $pCO_2$ was then estimated from measurements of dissolved inorganic carbon (DIC) and total alkalinity (ALK), together with sea surface temperature (SST), sea surface salinity (SSS), silicate, and phosphate concentrations. Monthly means were obtained by averaging all available estimates within each month. This approach ensures a consistent and physically based estimation of $pCO_2$ at the BAT site.

Table R3 (Table S3 in supplement section S3). List of selected independent test stations with long-term observations.

| Station | Coordinates | Time range | Number of samples | URL |
|---------|-------------|------------|-------------------|-----|
| BAT | 31.67°N, 295.83°E | 10/1991-6/2022 | 324 | https://bios.asu.edu/bats |
| HOT | 22.75°N, 202°E | 10/1988-12/2023 | 325 | https://hahana.soest.hawaii.edu/hot/hotco2 |
| ESTOC | 29.07°N, 344.17°E | 10/1995-11/2009 | 115 | https://www.ncei.noaa.gov/access/ocean-carbon-acidification-data-system/oceans/Coastal/ESTOC.html |
| CCE1 | 33.50°N, 237.50°E | 11/2008-12/2023 | 144 | https://www.ncei.noaa.gov/access/ocean-carbon-acidification-data-system/oceans/Moorings/Pacific.html |
| TAO | -0.51°N, 189.98°E | 2/2010-8/2016 | 45 | https://www.ncei.noaa.gov/access/ocean-carbon-acidification-data-system/oceans/Moori |

| | | | | ngs/Pacific.html |
|---|---|---|---|---|
| BOBOA | 15°N, 90°E | 11/2013-11/2018 | 53 | https://www.ncei.noaa.gov/access/ocean-car bon-acidification-data-system/oceans/Moori ngs/Indian.html |
| Papa | 50.13°N, 215.17°E | 6/2007-4/2023 | 168 | https://www.pmel.noaa.gov/co2/story/Papa |
| Iceland | 68°N, 347.40°E | 2/1985-11/2021 | 158 | https://www.ncei.noaa.gov/access/ocean-car bon-acidification-data-system/oceans/Moori ngs/Atlantic.html |
| Irminger | 64.30°N, 332°E | 3/1983-11/2012 | 99 | https://www.ncei.noaa.gov/access/ocean-car bon-acidification-data-system/oceans/Moori ngs/Atlantic.html |

6. The manuscript states that SJTU-AViT outputs were interpolated to the spatiotemporal locations of SOCAT for comparison, but it does not specify the interpolation method used (e.g., bilinear, nearest neighbor, or other), nor whether any temporal or spatial smoothing was applied. The authors should provide these details. For example, "When comparing with SOCAT, model values were interpolated to observation locations using bilinear interpolation in space and linear interpolation in time."

The Surface Ocean $CO_2$ Atlas (SOCAT) provides two forms of data: synthesis files and gridded (binned) products. The gridded SOCAT product is generated by interpolating individual observations onto a regular grid with a spatial resolution of 1° × 1° and a monthly temporal resolution. Only SOCAT observations with quality control (QC) flags of A–D and WOCE flags of 2 are included in this product. The arithmetic mean is first calculated for each cruise passing through a given grid cell, and these cruise-level means are then averaged to obtain the final gridded value. The resulting product provides fields with valid values in grid cells and months where observations are available, while grid cells without observational coverage are assigned NaN values.

We used the gridded SOCAT product, which share the same longitude and latitude grid as SJTU-AViT data. Therefore, there is no need for additional spatial or temporal interpolation. To account for gaps in the SOCAT data (NaNs), we mask the corresponding reconstructed values at the same grid–time points before computing any statistics. This ensures that all comparisons are performed only where SOCAT provides valid data. For the independent test at long-term stations, we use bilinear spatial interpolation to extract the reconstructed values at the corresponding station locations. This approach allows us to account for the surrounding grid cell information rather than relying solely on the nearest neighbor, thereby providing a smoother and more representative estimate of $spCO_2$ at the station sites.

To make this clearer, we have revised the section 2.3 (lines 212-220) to explicitly state our comparison procedure. The added text reads :

*"For comparison with SOCAT, we used the monthly 1° gridded SOCAT product and evaluated our SJTU-AViT reconstruction on the same grid, without applying any additional spatial interpolation. Reconstructed values were masked where SOCAT is*

*missing, and all skill metrics were computed only at grid-time points with valid SOCAT data. For the independent test at long-term stations, reconstructed values were extracted at the corresponding station locations using bilinear spatial interpolation, which incorporates information from surrounding grid cells to provide smoother and more representative estimates, and skill metrics were subsequently computed to evaluate model performance. Detailed information for these stations, including their names, geographic locations, observation periods, number of samples, and data sources, is provided in supplement Table S3, and their locations are shown in supplement Fig. S2 to facilitate visual interpretation."* .

7. The temporal coverage of Chl-a spans 1997–2022, whereas the product extends from 1982 to 2023. It is recommended to clarify how the periods prior to 1997 and for 2023 were handled (e.g., climatology, interpolation, gap-filling, or inference from other variables) to avoid any misunderstanding that the time spans are fully consistent.

The temporal coverage of the Chl-a dataset spans 1997-2022. For periods prior to 1997 and for 2023, we applied a climatology derived from the 1997-2022 record. We recognize that the use of a climatological mean does not capture interannual variability and may introduce a slight bias into the reconstruction. However, this effect is expected to be minor, while inferring Chl-a from other variables or extrapolating beyond the observational record could introduce substantially larger uncertainties. The use of a climatology therefore represents a pragmatic balance between competing sources of uncertainty, ensuring a stable and physically reasonable baseline. A similar approach has been adopted in other surface ocean $pCO_2$ reconstruction efforts, including Landschützer et al. (2016) and Gregor et al., (2021).

We have clarified this methodological choice in the revised manuscript (section 2.1, lines 115-118) to avoid any misunderstanding regarding the temporal consistency of the input variables, now as *"Chl-a data were derived from the European Space Agency Climate Change Initiative (ESA CCI) Ocean Colour (version 5.0) dataset, spanning 1997 to 2022 with daily resolution and a spatial resolution of 4 km (Jackson et al., 2017). For periods prior to 1997 and for 2023, we employed a climatology computed from the 1997-2022 Chl-a record to ensure full temporal coverage.".*

8. The manuscript employs a 2° monthly climatological MLD (WOCE). It is recommended to explain why a climatological mean was used instead of incorporating interannual and monthly variability, as this choice may affect the representation of temporal dynamics.

One of the most widely used and high-quality MLD dataset is the 2° global climatological MLD product by de Boyer et al. (2004), based on global observed temperature and salinity profiles. It accurately represents the climatological mean and provides robust physical constraints for long-term, large-scale $spCO_2$ reconstruction. During 1982-2023, high-resolution, continuous interannual MLD data are lacking globally. We acknowledge that climatological MLD does not capture interannual

variability, which may slightly underestimate spCO$_2$ interannual variability. However, introducing time-varying MLD from numerical ocean models could potentially introduce additional uncertainties, potentially larger than the bias introduced by using a climatology. Consequently, the use of a climatological MLD represents a pragmatic balance between different sources of uncertainty, ensuring a stable and reliable baseline for identifying large-scale, long-term patterns. A similar choice has been adopted in other ocean pCO$_2$ reconstruction, including Landschützer et al., (2016) and Gregor et al., (2021).

This limitation has been explicitly stated in the revised manuscript (section 4, lines 638-642), now as *"It should be noted that the climatological MLD used in this study cannot capture interannual or monthly variability, which may slightly underestimate local or short-term impacts on spCO$_2$. Nevertheless, it provides adequate physical constraints for reconstructing long-term and large-scale spatiotemporal patterns. Future work will explore incorporating high-quality time-varying MLD data as it becomes available to improve model fidelity at regional and seasonal scales."*.

**Minor comments:**

1. Please clarify whether the input data were standardized during the model training process, and specify the method used (e.g., variable-wise mean–variance normalization, min–max scaling, or other approaches).
During the model training process, the input data were standardized using variable-wise mean–variance normalization. Detailed descriptions of this procedure have been added in section 2.2 (lines 153-154) of the manuscript, now as *"These input variables are standardized using variable-wise mean-variance normalization and formatted into a multi-channel input to ensure feature extraction occurs on a unified scale."*.

2. Please clearly indicate the flux sign convention in the caption of Figure 13, for example: "Negative = ocean uptake (sink), Positive = release to the atmosphere (source)." Ensure that this convention is consistent with the main text, equations, and color bar.
Done as suggested. We have updated the figure caption (now is Fig. 14) to clearly indicate the flux sign convention (lines 576-577): *"Negative = ocean uptake (sink), Positive = release to the atmosphere (source)."* In addition, we have thoroughly checked the main text, relevant equations, and color bar to ensure that this convention is applied consistently throughout the manuscript.

3. In supplement Figure S6, the legend is labeled as "spCO2" which should be "spCO$_2$". Please ensure consistency of the symbol and formatting throughout the manuscript (e.g., uniformly using the subscript form "spCO$_2$" instead of "spCO2") and apply the same convention across all figures, captions, and text. In addition, please indicate the appropriate units (e.g., μatm) where relevant to avoid confusion.
Done as suggested. We have corrected the legend in supplement Figure S6 from "spCO2" to "spCO$_2$", as followed. In addition, we have carefully reviewed the entire

manuscript to ensure that the subscript formatting for "CO₂" is used consistently across all figures, captions, and text. We have also added the appropriate units (e.g., µatm) where relevant to avoid any potential confusion.

[Figure]

Figure R4. (Figure S6 in supplement section S4). Spatial distribution of standard deviation in interannual time scale of reconstructed spCO₂ at multiple data products from 1985 to 2018. All the panels show the standard deviation of residuals after removing long-term trends and seasonal cycles. The color scale represents the magnitude of variability in spCO₂, with higher values (red) indicating greater variability.

4. Line 113 — abbreviation usage: The term "Sea surface salinity (SSS)" repeats a definition already given earlier. Please use the abbreviation SSS here. A full-text check is recommended to correct similar inconsistencies.

Done as suggested. At the revised text (now is line 114), we have replaced "Sea surface salinity (SSS)" with the abbreviation SSS to avoid redundant definitions. In addition, we conducted a comprehensive review of similar cases throughout the manuscript to ensure that abbreviations are fully defined upon first appearance and consistently used thereafter.

5. Regarding xCO₂ (MBL), please clarify how the meridional band product was mapped onto the 1° × 1° grid (e.g., through band replication, interpolation, or another approach). Providing this detail would improve the transparency of the data processing procedure.

In our study, the meridional band xCO₂ (MBL) product, which is provided at discrete latitude bands, was mapped onto the 1° × 1° global grid using a two-step procedure:

(1) Latitudinal Interpolation: For each time step, the original xCO₂ values at discrete latitude bands were interpolated to the model's target latitudes using one-dimensional linear interpolation along the meridional direction (implemented with MATLAB's interp1 function). This ensures a smooth transition of xCO₂ values between the original latitude bands.

(2) Longitudinal Replication: Because the original xCO₂ product does not contain longitudinal variations, the interpolated latitudinal profile was replicated along all longitudes to produce a complete 2D global field at 1° × 1° resolution. This approach

preserves the meridional gradient while assuming longitudinal uniformity, consistent with the original dataset.

This mapping procedure was applied to all time steps from 1982 to 2023. These details have now been explicitly added to the section 2.1 (lines 122-124) to improve the transparency and reproducibility of the data processing, now as *"In this study, the meridional band product was mapped onto the model's 1° × 1° global grid using latitudinal interpolation and longitudinal replication, generating continuous 2D fields suitable for model simulations."*

6. In the Methods section, please specify the training setup, including the maximum number of epochs and/or the early stopping patience (e.g., "trained for up to 200 epochs with early stopping, patience = 20"), to improve the reproducibility of the approach.

In our study, the ViT-based model was trained for a maximum of 200 epochs with early stopping applied, using a patience of 10 epochs. This means that training would terminate if the validation loss did not improve for 10 consecutive epochs. Each training epoch required roughly 10 minutes. These details have now been explicitly added to section 2.2 (lines 176-178) to improve the clarity and reproducibility of our approach, now as *"The ViT-based model contains approximately 115 million parameters and was trained in parallel on eight NVIDIA RTX 4090 GPUs for up to 200 epochs with early stopping (patience = 10); each training epoch required roughly 10 minutes."*

7. It is recommended to indicate the sample size for each data point or category in Figure 3, allowing readers to more clearly understand the data coverage and the reliability of the statistics.

Following the comment, we have indicated the sample size for each data station in revised figure (now is Fig. 4). The addition is shown in the figure below.

[Figure]

Figure R5. (Figure 4 in main text). Independent test of $spCO_2$ variability between SJTU-AViT and in situ observations at different stations. These in situ data are independent data and are not used to train the model. The station description and location refer to supplement section S2 and Fig. S2. The $spCO_2$ in SJTU-AViT is interpolated to match the station locations and times in the comparison. For each panel, the number of samples (N), the mean bias error (MBE), root mean square error (RMSE), and correlation coefficient ($R^2$) between the reconstructed and observed $spCO_2$ are displayed. The dashed and solid lines show the linear trend of SJTU-AViT and in situ data.

8. It is recommended to review the entire manuscript and ensure that all instances of "CO2" use a subscript for the number 2, maintaining consistency and adhering to scientific writing conventions.
Done as suggested. We have carefully reviewed the entire manuscript and ensured that all instances of "CO2" now use a subscript for the number 2, maintaining consistency throughout the text and adhering to standard scientific writing conventions.

9. Line 31: In the abstract, change "This study not only provide…" to "This study not only provides…". It is recommended to review the entire manuscript for program errors.
Done as suggested. We have corrected the sentence in the abstract (now is line 32) to *"This study not only provides…"*. In addition, we have carefully reviewed the entire manuscript to identify and correct similar grammatical errors, ensuring the accuracy

and readability of the text.

10. Line 31: In the abstract, change "Earth system" to "Earth-system" when used as a compound adjective for clarity
Done as suggested. We have revised the abstract (now is line 34) to change "Earth system" to "Earth-system" when used as a compound adjective.

11. Line 170: It is recommended to revise the sentence to: "The ViT-based model contains approximately 115 million parameters and was trained in parallel on eight NVIDIA RTX 4090 GPUs; each training epoch required roughly 10 minutes."
Done as suggested. Incorporating the suggestion from minor comment 6, we have revised the sentence (now is lines 176-178) to: *"The ViT-based model contains approximately 115 million parameters and was trained in parallel on eight NVIDIA RTX 4090 GPUs for up to 200 epochs with early stopping (patience = 10); each training epoch required roughly 10 minutes."*

12. Line 260: It is recommended to revise the sentence to: "Most predicted values lie close to the 1:1 line, particularly within the climatologically common spCO$_2$ range (300–420 µatm), as indicated by the high-density regions in Fig. 2."
Done as suggested. We have revised the sentence (now is lines 287-289) to: *"Most predicted values lie close to the 1:1 line, particularly within the climatologically common spCO$_2$ range (300-420 µatm), as indicated by the high-density regions in Fig. 3."*

13. Line 312: Ensure there is a space before "µatm," e.g., "-12 µatm to +10 µatm."
Done as suggested. We have corrected the formatting issue by adding a space before "µatm" (e.g., "-12 µatm to +10 µatm") in the revised text (now is line 340). We have also carefully checked the entire manuscript to ensure consistent formatting of units throughout.

14. It is recommended to standardize the number of decimal places throughout the manuscript (e.g., consistently using two or three decimal places).
We have carefully checked all numerical values reported in the manuscript and standardized the format to retain two decimal places throughout. We note, however, that for prescribed data such as Niño 3.4 and the scaling factor (set as 0.251), no modification was made.

15. For the air–sea flux calculation, the parameterization of Wanninkhof (2014) requires the Schmidt number, wind speed source, and resolution (you used ERA5). It is recommended to specify in section 2.4 the temporal and spatial resolution of ERA5 and the formula or reference used for computing the Schmidt number.
Done as suggested. In our study, the air-sea CO$_2$ flux was calculated using the parameterization of Wanninkhof (2014). The wind speed data were sourced from the ERA5 reanalysis, with a 6-hourly temporal resolution covering 1982-2023 and a

horizontal spatial resolution of 1°.

The gas transfer velocity of $CO_2$ ($k_w$) is computed using a quadratic dependence on wind speed:

$$k_w = 0.251 \cdot (Sc/660)^{-\frac{1}{2}} \cdot u^2$$

where $u$ is the wind speed at 10 m above the sea surface, and $Sc$ is the Schmidt number of $CO_2$ in seawater. The Schmidt number is calculated from the temperature-dependent empirical formula:

$$Sc = 2116.8 - 136.25 \cdot T + 4.7353 \cdot T^2 - 0.092307 \cdot T^3 + 0.0007555 \cdot T^4$$

where T is the sea surface temperature in °C. This formulation accounts for the effect of sea surface temperature on $CO_2$ diffusivity in seawater.

These details, including the information on ERA5 data and the reference used for computing the Schmidt number, have been added to section 2.4 (lines 257-259) of the manuscript. They are now described as: *"The Schmidt number (Sc) required in this formulation is calculated following the temperature-dependent empirical formula provided by Wanninkhof (2014). The wind speed data is sourced from ERA5, with a 6-hourly temporal resolution spanning 1982-2023 and a 1° spatial resolution."*

16. Table 1: It is recommended to change "12 month" to "12-month."
Done as suggested. This correction has been applied in Table 1 and carefully reviewed throughout the manuscript to ensure consistency.

17. In the Abstract, it is stated that the model shows a correlation of 0.81 with the Niño 3.4 index, whereas section 3.4 reports a correlation of –0.81. This inconsistency in the sign of the correlation may confuse readers. Please verify the original calculation and ensure that the values and their signs are reported consistently throughout the manuscript.
Done as suggested. The value reported in the Abstract (0.81) was a typo and the correct correlation coefficient is -0.81. This has been corrected in the revised manuscript (Abstract, line 29).

References not in manuscript:

Dickson, A. G.: Standard potential of the reaction: , and the standard acidity constant of the ion HSO in synthetic sea water from 273.15 to 318.15 K, J. Chem. Thermodyn., 22, 113-127, https://doi.org/10.1016/0021-9614(90)90074-Z, 1990.

Humphreys, M. P., Lewis, E. R., Sharp, J. D., and Pierrot, D.: PyCO2SYS v1.8: marine carbonate system calculations in Python, Geosci. Model Dev., 15, 15-43, https://doi.org/10.5194/gmd-15-15-2022, 2022.

Waters, J. F. and Millero, F. J.: The free proton concentration scale for seawater pH, Mar. Chem., 149, 8-22, https://doi.org/10.1016/j.marchem.2012.11.003, 2013.

Perez, F. F. and Fraga, F.: Association constant of fluoride and hydrogen ions in seawater, Mar. Chem., 21, 161-168,

https://doi.org/10.1016/0304-4203(87)90036-3, 1987.

---

## Author Comment (AC2)

**Response to Referee #2**

**General comments:**

Zhang et al. present a global monthly surface ocean $pCO_2$ dataset (SJTU-AViT) and corresponding air-sea $CO_2$ fluxes spanning 1982-2023 at 1° resolution, developed using a Vision Transformer-based deep learning model. The approach combines SOCAT observation, and observations of climate data with multiple ocean biogeochemical models and incorporates physical-biogeochemical constraints. The authors show that their product successfully captures the spatial and temporal variations of observed $pCO_2$ patterns, from seasonal cycles to interannual variability. The product shows more realistic small-scale spatial variability and temporal interannual variability than previous $pCO_2$ products. The resolved air-sea $CO_2$ fluxes agree with other estimates based on $pCO_2$ observations. The paper is well written, the methodology is robust, and the line of thought is mostly clear to me. I only have minor comments regarding some of the technical details and presentation.

We thank the reviewer for the helpful and constructive feedback. We have revised the manuscript to address all of these comments. Overall, the reviewer's main concerns focused on the transparency and robustness of the model training strategy, the contribution of physical-biogeochemical constraints, the adequacy of uncertainty estimation method, and several issues related to data processing and presentation. In response to these concerns, we have made the following revisions.

- Clarify and validate the two-stage training framework, and quantify the contributions of its components. We elaborated the physical motivations for CMIP6 pre-training, MOM6 constraints, and SOCAT fine-tuning, and added ablation experiments to demonstrate their respective roles in improving convergence, generalization, and accuracy (major comment #1).
- Revise the uncertainty estimation framework. We replaced the observation-dependent $u_{map}$ with an algorithm-based uncertainty estimate ($u_{algorithm}$) derived from synthetic sampling experiments, and integrated the complete workflow and quantitative results into the Methods and Results sections (major comment #2).
- Enhance diagnostic analyses and visualization. We improved the calculation of seasonal variability by applying linear detrending prior to analysis, and added seasonal-phase diagnostics and peak–minimum month difference maps (minor comments #7, #9).
- Revise minor edits and clarifications (minor comments #1-11).

Please see our detailed point-by-point responses to each comment below.

**Major comments:**

1. The description of methodology is overall complete. However, certain technical details are still missing. It is not clear how pre-training on CMIP6 models contributes to the final model. It is not clear what the fine-tuning of MOM6 really does. Are your results sensitive to the choice of CMIP6 models and the fine-tuning? How do SOCAT

data fold into your refinement? For the physical-biogeochemical constraints, are you only using what is derived from MOM6, or also from CMIP6 models as well? How are your results, particularly on the seasonal cycle, impacted by these physical-biogeochemical constraints? In other words, if you exclude these constraints, how is the representation of the seasonal $pCO_2$ cycle affected?

We thank the reviewer for raising this comprehensive question. We have structured our response into six corresponding parts for clarity. The revisions include 5 ablation experiments, with summary findings presented in the main text (section 4, lines 601-612) and full experimental details reported in the supplement (section S5.2-S5.6).

(1) How pre-training on CMIP6 models contributes to the final model?

To quantitatively assess the impact of CMIP6-based pretraining on the reconstruction, we conducted two controlled experiments that were identical in all settings except for the use of CMIP6 pretraining.

   (a) Test 1 (with CMIP6 pretraining): The model was first pretrained on CMIP6 simulation outputs, allowing it to learn from CMIP6 model results. It was then jointly fine-tuned using MOM6 and SOCAT observational data.

   (b) Test 2 (without CMIP6 pretraining): Under the same conditions, the model relied solely on MOM6 and SOCAT data.

   The ablation experiments reveal a substantial impact of CMIP6 pretraining on the results. When pretrained on CMIP6 (Test 1), the model achieved an RMSE of 7.44 µatm on the validation set. Without CMIP6 pretraining (Test 2), RMSE increased to 17.13 µatm. Thus, CMIP6 pretraining reduced RMSE by 9.69 µatm, corresponding to a relative decrease of approximately 56.57%. The spatial map (Fig. R1) indicates that the largest improvements occur in regions with sparse observations (particularly at high latitudes) and areas with pronounced low-frequency or interannual variability.

   CMIP6 pretraining provides the model with a physically meaningful initialization. By learning from temporally and spatially complete simulation fields, the model can first capture large-scale spatial patterns and low-frequency signals, enabling faster convergence during fine-tuning, reducing overfitting in observation-sparse regions, and achieving better generalization at interannual scales. Although CMIP6 simulations may contain biases, these are effectively corrected during the subsequent fine-tuning with MOM6 and SOCAT, ensuring the final reconstruction remains consistent with observations. The substantial RMSE improvement (a reduction of 9.69 µatm, ~56.57%) demonstrates that this two-stage training strategy achieves an optimal balance between physical consistency and empirical accuracy.

[Figure]

Figure R1 (Figure S9 in supplement section S5). Impact of CMIP6 pre-training on reconstructed spCO$_2$ fields. (a) Test 1 (with CMIP6 pretraining): CMIP6 pre-training followed by MOM6 & SOCAT fine-tuning; (b) Test 2 (without CMIP6 pretraining): no CMIP6 pre-training, trained only on MOM6 & SOCAT. Inclusion of CMIP6 pre-training reduces validation RMSE by 9.69 µatm (~56.57% relative reduction), justifying the two-stage training strategy.

(2) What the fine-tuning of MOM6 really does?

To assess the role of MOM6 fine-tuning in our reconstruction framework, we designed two comparative experiments while keeping all other model settings identical:

   (a) Test 1 (with MOM6 in fine-tuning): The model was first pretrained on CMIP6 outputs and then fine-tuned using both MOM6 simulation outputs and SOCAT observations. MOM6 provides continuous, physically consistent global fields, while SOCAT supplies essential observational constraints.

   (b) Test 2 (without MOM6 in fine-tuning): The model was pretrained on CMIP6 data as in Test 1 but fine-tuned solely with SOCAT observations, without incorporating MOM6 outputs.

   The fine-tuning strategy that included MOM6 data (Test 1) achieved a validation RMSE of 7.44 µatm. In contrast, excluding MOM6 during fine-tuning (Test 2) resulted in a substantially higher RMSE of 12.27 µatm. Thus, incorporating MOM6 during fine-tuning reduced RMSE by 4.83 µatm, corresponding to a relative decrease of approximately 39.36%. The spatial map (Fig. R2) indicates that the largest improvements occur in regions with sparse observations, particularly at high latitudes, and in areas with pronounced low-frequency or interannual spCO$_2$ variability, highlighting the crucial role of MOM6 in enhancing reconstruction accuracy.

   In our framework, MOM6 outputs are incorporated alongside SOCAT observations during the fine-tuning stage. SOCAT provides the essential observational constraint, but its spatial and temporal coverage is sparse and uneven. MOM6 complements this by supplying continuous global fields that embed large-scale physical consistency, thereby stabilizing the training process and enhancing generalization, particularly in data-poor regions. Mechanistically, MOM6 fine-tuning serves three key functions: (i) it exposes the network to continuous, globally coherent background fields (e.g.,

large-scale gradients, seasonal cycles, and interannual variability), thereby reducing overfitting to the sparse and uneven SOCAT distribution; (ii) it aligns model weights with physically plausible oceanographic relationships, mitigating the direct transfer of structural biases from heterogeneous CMIP6 pre-training and avoiding abrupt or unrealistic weight corrections during SOCAT anchoring; (iii) it supplies realistic background variability, enabling the model to learn coherent patterns prior to adjustment with pointwise observations, which strengthens generalization in data-limited regions. In summary, MOM6 fine-tuning functions as a physically consistent bridge between synthetic CMIP6 pre-training and sparse SOCAT observations, significantly improving the stability, robustness, and reliability of the reconstruction, especially in regions with limited observational coverage.

[Figure]

Figure R2 (Figure S10 in supplement section S5). Impact of MOM6 fine-tuning on reconstructed $spCO_2$ fields. (a) Test 1 (with MOM6 in fine-tuning): CMIP6 pre-training followed by MOM6 & SOCAT fine-tuning; (b) Test 2 (without MOM6 in fine-tuning): CMIP6 pre-training, fine-tuning only on SOCAT. Inclusion of MOM6 fine-tuning reduces validation RMSE by 4.83 µatm (~39.36% relative reduction), highlighting the crucial role of MOM6 in enhancing reconstruction accuracy.

(3) Are your results sensitive to the choice of CMIP6 models and the fine-tuning?
To assess the sensitivity of our reconstruction to the choice of CMIP6 models and the fine-tuning strategy, we conducted two comparative pre-training experiments while keeping all other model settings identical:

   (a) Test 1 (3-model CMIP6 pre-training): The model was pre-trained on a subset of three CMIP6 simulations (GFDL-ESM4, NorESM2-LM, NorESM2-MM) and then fine-tuned with the same MOM6 and SOCAT data.

   (b) Test 2 (4-model CMIP6 pre-training): The model was pre-trained on a different subset of four CMIP6 simulations (CESM2, CESM2-FV2, CESM2-WACCM, CESM2-WACCM-FV2) and fine-tuned using the same MOM6 and SOCAT data.

   The ViT reconstruction using the 3-model subset (Test 1) achieved a validation RMSE of 10.48 µatm, while the 4-model subset (Test 2) yielded a slightly lower RMSE of 9.54 µatm. Both are higher than the RMSE obtained using all seven CMIP6 models (7.44 µatm), indicating that the total amount of pre-training data can influence reconstruction performance. Nevertheless, the difference between the two subsets is

small (RMSE difference of 0.94 µatm, ~8.97%), and deviations from the 7-model pre-training result are modest (~2-3 µatm).

Overall, these results indicate that, as long as multiple CMIP6 models are included to capture diverse large-scale oceanic patterns, the reconstruction is largely robust to the specific choice of pre-training models. The two-stage training framework effectively stabilizes reconstruction performance, corrects model-specific biases, and reliably integrates observational information. To further strengthen robustness, CMIP6 models were carefully selected based on the evaluation framework of Liao et al. (2021), ensuring that the chosen models accurately represent key oceanic carbon dynamics. Through multi-model pre-training combined with carefully designed fine-tuning strategies, our approach maintains stable and reliable reconstruction performance, effectively capturing large-scale patterns, low-frequency variability, and regional details across different spatial and temporal scales.

The reconstruction results are robust to reasonable systematic changes in key fine-tuning hyperparameters (such as learning rate, batch size, patch size, and Transformer block number) though extreme changes (e.g., reducing Transformer blocks from 10 to 5) can substantially affect performance. Fine-tuning data are crucial: MOM6 provides physically consistent global fields to stabilize training and enhance generalization (see response 1.2), while SOCAT observations correct local and regional biases (see response 1.4), together ensuring stable, reliable, and physically coherent spCO$_2$ reconstructions across both well-observed and data-sparse regions.

[Figure]

Figure R3 (Figure S11 in supplement section S5). The sensitivity of reconstructed spCO$_2$ fields to the choice of CMIP6 models. (a) Test 1 (3-model CMIP6 pre-training): three CMIP6 simulations (GFDL-ESM4, NorESM2-LM, NorESM2-MM) pre-training followed by MOM6 & SOCAT fine-tuning; (b) Test 2 (4-model CMIP6 pre-training): four CMIP6 simulations (CESM2, CESM2-FV2, CESM2-WACCM, CESM2-WACCM-FV2) pre-training followed by MOM6 & SOCAT fine-tuning.

(4) How do SOCAT data fold into your refinement?
To evaluate the role of SOCAT observations in the fine-tuning stage, we designed two comparative experiments while keeping all other model settings identical:

(a) Test 1 (with SOCAT in fine-tuning): The model, pretrained on CMIP6 and optionally fine-tuned with MOM6 fields, was further fine-tuned using SOCAT in situ

pCO₂ observations. SOCAT provides high-quality pointwise constraints that correct model biases and ensure alignment with real-world ocean conditions.

(b) Test 2 (without SOCAT in fine-tuning): The same pretrained model was fine-tuned without using SOCAT data, relying solely on MOM6 fields for spatial coverage and physical consistency.

Incorporating SOCAT observations during fine-tuning (Test 1) yielded a validation RMSE of 7.44 µatm. In contrast, excluding SOCAT (Test 2) resulted in a dramatically higher RMSE of 26.87 µatm. Thus, the inclusion of SOCAT reduced RMSE by 19.43 µatm, corresponding to a relative decrease of approximately 72.31%. This large improvement demonstrates the critical role of SOCAT observations in aligning the reconstructed spCO₂ field with real-world measurements.

SOCAT data act as a supervisory signal that corrects local and regional biases in the model, ensuring the fine-tuned reconstruction reproduces observed variability while retaining large-scale spatiotemporal patterns learned during CMIP6 pretraining and MOM6 fine-tuning. Without SOCAT, the model cannot accurately capture local pCO₂ variations, leading to substantial errors. Proper integration of SOCAT with MOM6 fields balances the influence of sparse observational points and physically consistent background patterns, enhancing overall predictive skill, particularly in regions with limited observations.

[Figure]

Figure R4 (Figure S12 in supplement section S5). Impact of SOCAT observations on the fine-tuning of the reconstructed spCO₂ field. (a) Test 1 (with SOCAT in fine-tuning): CMIP6 pre-training followed by MOM6 & SOCAT fine-tuning; (b) Test 2 (without SOCAT in fine-tuning): CMIP6 pre-training, fine-tuning only on MOM6. Inclusion of SOCAT observations reduces validation RMSE by 19.43 µatm (~72.31% relative reduction), demonstrating the pivotal role of SOCAT in achieving accurate spCO₂ reconstruction.

(5) For the physical-biogeochemical constraints, are you only using what is derived from MOM6, or also from CMIP6 models as well?

In our study, the physical-biogeochemical constraints incorporated in the ViT model are derived exclusively from MOM6 simulations. MOM6 provides high-resolution, rigorously validated ocean-driven fields that more accurately represent conditions relevant to the surface carbon system (Stock et al., 2020; Liao et al., 2020).

Evaluations such as those by Liao et al. (2021) indicate that CMIP6 model outputs contain relatively large biases in space and time, which could reduce the reliability of any constraints derived directly from CMIP6. Therefore, MOM6 is used as the sole source for physical-biogeochemical constraints to ensure accuracy, consistency, and physical realism in the model refinement stage. In addition, we have conducted comparison experiments in the manuscript between models trained with and without these constraints, demonstrating that incorporating MOM6-derived information significantly improves predictive skill in data-sparse regions and high-latitude oceans.

(6) How are your results, particularly on the seasonal cycle, impacted by these physical-biogeochemical constraints? In other words, if you exclude these constraints, how is the representation of the seasonal $pCO_2$ cycle affected?

To assess the impact of MOM6-derived physical-biogeochemical constraints on the seasonal cycle of $spCO_2$, we conducted two comparative experiments while keeping all other model settings identical:

   (a) Test 1 (with physical-biogeochemical constraints): The SJTU-AViT model reconstruction incorporated MOM6-derived constraints during training, enforcing physically and biogeochemically plausible relationships among environmental variables.

   (b) Test 2 (without physical-biogeochemical constraints): The SJTU-AViT reconstruction excluded these constraints, allowing the model to rely solely on observational and CMIP6-derived information.

   The constraints systematically improve model performance across all seasons (Fig. R5), as reflected in reduced RMSE values: MAM decreases from 11.66 to 11.35 µatm (~2.66%), JJA from 12.31 to 11.93 µatm (~3.09%), SON from 13.67 to 12.51 µatm (~8.49%), and DJF from 10.32 to 10.18 µatm (~1.36%). On average, the inclusion of constraints reduces RMSE by ~3.90% across the four seasons.

   These improvements are systematic and physically meaningful rather than random fluctuations. The MOM6-derived constraints anchor the model to physically and biogeochemically plausible relationships, enhancing the accuracy and robustness of the seasonal $spCO_2$ representation. The constraints are particularly effective in regions with sparse observational coverage, where purely data-driven reconstructions may be prone to larger errors. Overall, the results demonstrate that including physical-biogeochemical constraints play a substantial and reliable role in improving the seasonal cycle representation of $spCO_2$, rather than merely introducing stochastic or localized enhancements.

[Figure]

Figure R5 (Figure S13 in supplement section S5). Seasonal comparison of SJTU-AViT spCO$_2$ means and RMSE with and without physical-biogeochemical constraints. (a-d) Test 1 (with physical-biogeochemical constraints): seasonal mean spCO$_2$ from SJTU-AViT with physical-biogeochemical constraints for MAM (March-May), JJA (June-August), SON (September-November), and DJF (December-February). (e-h) Test 2 (without physical-biogeochemical constraints): seasonal mean spCO$_2$ from SJTU-AViT without constraints. (i-l) Test 1 (with physical-biogeochemical constraints): seasonal RMSE of spCO$_2$ between SJTU-AViT and SOCAT with constraints. (m-p) Test 2 (without physical-biogeochemical constraints): seasonal RMSE of spCO$_2$ between SJTU-AViT and SOCAT without constraints. For RMSE calculations, SJTU-AViT spCO$_2$ was interpolated to SOCAT observation locations and times.

The results are summarized in the main text (lines 601-612) as *"In addition, we evaluated the contributions of CMIP6 pre-training, MOM6 fine-tuning, SOCAT observations, and MOM6-derived physical-biogeochemical constraints within the SJTU-AViT framework. CMIP6 pre-training substantially improved model initialization and skill, reducing validation RMSE by ~56.57% versus random initialization by supplying large-scale structure and low-frequency variability. MOM6 fine-tuning further stabilized the model—especially in observation-sparse regions—lowering RMSE by ~39.36% and enforcing physically plausible relationships. Including SOCAT during fine-tuning was critical for local and regional accuracy, reducing RMSE by ~72.31% through high-quality pointwise constraints. Sensitivity tests indicate the reconstruction is largely robust to the specific choice of CMIP6 pre-training subsets, provided multiple models are used to capture diverse large-scale patterns. Finally, adding MOM6-derived physical constraints improved overall performance (MAE from 7.15 to 5.95 µatm) and reduced seasonal RMSE by*

*1.36-8.49%, with the largest gains in high-latitude and data-sparse regions. Collectively, these results confirm that CMIP6 pre-training followed by MOM6- and SOCAT-constrained fine-tuning with physically informed constraints yields a robust, reliable, and physically consistent reconstruction of spCO$_2$ across spatial and temporal scales.".*

2. The uncertainty quantification might benefit from more detail. For $u_{map}$, what if there are no observations in one grid? How do you then quantify $u_{map}$ there? Have you conducted an analysis on the spatial heterogeneity of the dominant source of uncertainty? In addition, I think it would be more appropriate to replace $u_{map}$ with "algorithm uncertainty." Perhaps this can be done by generating a large ensemble of spCO$_2$ Alternatively, this can be done by using synthetic data. You might consider subsampling SOCAT data from one of your models and then applying the ML model to subsampled model fields to generate an spCO$_2$ map. Then you can compare the absolute differences between pCO$_2$ from the ocean model and the ML reconstruction. We acknowledge that the traditional $u_{map}$ approach depends directly on observational coverage and may underestimate uncertainty in regions with sparse or missing SOCAT data. To address this limitation, we performed an additional experiment using synthetic data to provide a more robust estimate of algorithm uncertainty. Specifically, we used the RECCAP2 simulation from the Scott Doney group (hereafter SD data) as an independent reference "truth," which the ViT machine learning model had never seen before. The SD data were divided into two subsets:

● SD_SOCAT: SD outputs sampled at the spatiotemporal locations of SOCAT observations.
● SD_nonSOCAT: the remaining SD outputs.

Following our standard workflow (CMIP6 pretraining, MOM6 fine-tuning, and SD_SOCAT fine-tuning), we reconstructed spCO$_2$ and quantified three RMSE values:

(a) RMSE_SD_SOCAT = 5.58 µatm. This is bias at training locations, indicating good consistency with data the model has seen.

(b) RMSE_SD_nonSOCAT = 7.40 µatm. This is bias at independent validation points, demonstrating generalization to unseen data.

(c) RMSE_SD_all = 7.39 µatm. This is bias over the full SD dataset, reflecting the model's overall performance.

These results show that the training error is slightly lower, as expected, and the validation and overall errors are nearly identical. This indicates that the ViT model does not overfit and that its uncertainty estimates are robust across different spatial domains. The close agreement also demonstrates that algorithm uncertainty captures the spatial heterogeneity of errors, particularly in high-latitude or data-sparse regions where $u_{map}$ cannot be defined.

Based on this analysis, we adopt the RMSE_SD_all = 7.39 µatm as a quantitative measure of algorithm uncertainty ($u_{algorithm}$), and have updated the manuscript accordingly in section 2.5 (lines 273-274) and section 3.6 (lines 580-581). Specifically, it is now stated as: *"$u_{algorithm}$ is evaluated as the RMSE between the reconstructed and reference ocean model spCO$_2$ field."* (lines 273-274) and *"with the*

*dominant contribution arising from the algorithm uncertainty ($u_{algorithm}$), which reaches 7.39 µatm.”* (lines 580-581). The full experimental details have been reported in the supplement (section S5.7).

[Figure]

Figure R6 (Figure S14 in supplement section S5). Spatial distribution of RMSE (µatm) between the reconstructed $spCO_2$ field and the Scott Doney RECCAP2 simulation (SD data). (a) RMSE for the full SD dataset. (b) RMSE for the SD_SOCAT subset, i.e., SD data sampled at SOCAT observation locations and used in training. (c) RMSE for the SD_nonSOCAT subset, i.e., SD data at locations not sampled by SOCAT and reserved for independent validation. The mean RMSE value for each panel is indicated. The SD data is from Doney et al., (2009).

**Minor comments:**

1. L15-16: The statement that ocean surface partial pressure of $spCO_2$ directly determines the air-sea $CO_2$ flux is not exactly correct. It is the air-sea $pCO_2$ difference, which is modulated by surface wind speed and gas exchange velocity.
We agree that the original description was not accurate and have revised the corresponding sentence in the Abstract (lines 15-18) to read:*“The ocean plays a crucial role in regulating the global carbon cycle and mitigating climate change. Spatial and temporal variations of ocean surface partial pressure of $CO_2$ ($spCO_2$) influence the air-sea $CO_2$ flux through the difference between surface ocean and atmospheric $pCO_2$ ($\Delta pCO_2$), which is further modulated by surface wind speed and gas exchange velocity.”*

2. Introduction: Perhaps it is also worth mentioning that previous ML-interpolation of $pCO_2$ overly smooths the spatial patterns and interannual variability.
We have revised the Introduction accordingly (lines 67-69): *“Previous machine learning (ML)-based interpolations of $pCO_2$ may overly smooths the spatial patterns and interannual variability, which represents a potential limitation in capturing these features fully.”*

3. L195: Is the interpolation based on inverse distance weighted average? How do you deal with the fine-resolution time (i.e., not monthly average)?
A similar question was also raised by the other reviewer. To clarify, we directly used the monthly $1° \times 1°$ gridded product provided by the Surface Ocean $CO_2$ Atlas (SOCAT) for data construction. Therefore, no additional spatial interpolation was applied, and the temporal resolution is already monthly. To handle missing values, we

masked the corresponding reconstructed values at the same grid-time points before computing statistics, ensuring that all comparisons are made only where SOCAT provides valid data.

For independent validation at long-term stations, reconstructed values were extracted at the station locations using bilinear interpolation from the surrounding grid cells, rather than simply selecting the nearest grid cell. This approach yields smoother and more representative $spCO_2$ estimates. All datasets, including these station comparisons, were consistently processed as monthly averages, with no further temporal interpolation.

We have revised the section 2.3 (lines 212-220) accordingly. The new text reads:*"For comparison with SOCAT, we used the monthly 1° gridded SOCAT product and evaluated our SJTU-AViT reconstruction on the same grid, without applying any additional spatial interpolation. Reconstructed values were masked where SOCAT is missing, and all skill metrics were computed only at grid-time points with valid SOCAT data. For the independent test at long-term stations, reconstructed values were extracted at the corresponding station locations using bilinear spatial interpolation, which incorporates information from surrounding grid cells to provide smoother and more representative estimates, and skill metrics were subsequently computed to evaluate model performance. Detailed information for these stations, including their names, geographic locations, observation periods, number of samples, and data sources, is provided in supplement Table S3, and their locations are shown in supplement Fig. S2 to facilitate visual interpretation."*.

4. Figure 3: Systematic biases are clear at Iceland and Irminger, with SJTU-AViT underestimating the $pCO_2$. Any clues why?

These high-latitude regions are strongly influenced by processes such as seasonal sea-ice coverage and freshwater input from precipitation, which are not well captured in our machine learning model due to the lack of corresponding observational constraints. As a result, the model cannot fully resolve these $pCO_2$ variabilities, leading to the observed negative bias. This behavior is not unique to our product and similar biases have been reported in other reconstruction products under complex environmental conditions (e.g., Landschützer et al., 2016; Gregor et al., 2021). In future work, we plan to incorporate additional predictors, such as non-climatological mixed-layer depth (MLD), sea-ice coverage, precipitation, and chlorophyll, into the machine learning framework to improve reconstruction accuracy in high-latitude regions.

Modified sentence in the section 3.1 (lines 302-306): *"At the Irminger Sea and Iceland sites, the model exhibits large RMSE (35.24 and 21.82 µatm, respectively) and low correlations, with $R^2$ near zero. This suggests that the model has difficulty capturing rapid $spCO_2$ fluctuations or processes that are not well represented by the available input features. This discrepancy is likely due to high-latitude processes such as seasonal sea-ice variability and freshwater inputs, which are not fully represented in the current observational constraints."*

5. Figure 5: The negative bias would lead to an overestimation of global ocean $CO_2$ uptake through the bulk equation. Might be worth mentioning when you talk about the flux.

We have mentioned this potential bias in the updated manuscript in section 3.5. Specifically, while SJTU-AViT effectively reproduces the overall spatial patterns and mechanisms of air-sea $CO_2$ flux, negative sp$CO_2$ biases remain in certain high-latitude regions (now is Fig. 6). These biases probably result from underestimation of p$CO_2$ in areas affected by seasonal sea-ice variability, freshwater inputs, and other high-latitude processes that are not fully captured by observational constraints.

The revised text now reads (section 3.5, lines 559-563): *"While SJTU-AViT effectively reproduces the overall spatial patterns and mechanisms of air-sea $CO_2$ flux, Figure 6 indicates that negative sp$CO_2$ biases remain in certain high-latitude regions. The negative bias, likely associated with underrepresented high-latitude processes such as seasonal sea-ice variability and freshwater inputs, can lead to an overestimation of global ocean $CO_2$ uptake through the bulk equation and should be considered when interpreting the absolute flux magnitude."*.

6. Fig. 6b: Seems like the bias PDF is wider in certain years. Speculation?

It is noteworthy that the bias probability density function (PDF) exhibits interannual variability and even decadal trends (now is Fig. 7b). In the early years (1980s to mid-1990s), the bias distribution is relatively broad, reflecting larger uncertainties. This is probably attributable to the sparse SOCAT coverage during that period. Limited observational data constrained the model's ability to resolve local and temporal variations, leading to larger bias. Over time, as SOCAT coverage expanded, reconstruction accuracy improved in most regions. The bias distribution became narrower and more symmetric, with the PDF centered near zero, indicating reduced systematic bias.

In recent years, however, the bias range appears to increase. This widening is likely related to the extension of observational coverage into high-latitude, polar, and coastal regions, where conditions are more variable and extreme. In addition, recent ocean p$CO_2$ changes have exhibited enhanced seasonal and interannual variability, which the model may not fully capture, particularly under extreme or marginal conditions. These interpretations remain tentative, and more detailed analyses—such as targeted experiments and the incorporation of additional datasets—will be necessary to fully disentangle these drivers. We view this as an important avenue for future work and would welcome collaborations to further investigate these aspects.

This has been added to section 3.2 (lines 370-375) as *"However, we note that the absolute range of biases may increase in later years. This widening is likely due to a combination of factors, including the expansion of observational coverage to regions with more extreme or marginal conditions, which introduces a larger range of reconstructed values, as well as the enhanced seasonal and interannual variability that the model may not fully capture in some regions, leading to increased biases under local or extreme conditions. Overall, the temporal evolution of the bias distribution highlights both the influence of observational coverage and the*

*challenges in capturing high-frequency or extreme variations.".*

7. L369-372: The section title is on the seasonal cycle, but the first few sentences focus on variability at all time scales. Might consider moving this to a later section. Also, the trend should be removed beforehand in calculating STD in Fig. 7.

We carefully considered splitting this section into two parts (full variability and seasonal variability). However, doing so would result in very little content for the full variability part and lead to an imbalance between subsections. Therefore, in the revised manuscript we chose to keep the two components together but revised the section title to *"3.3 Evaluation of full spCO₂ variability and seasonal cycle"* (line 399), which more accurately reflects the content.

In addition, as suggested, we removed the long-term trend prior to calculating the standard deviation (STD) in the figure (now is Fig. 8 and Fig. S5). After detrending, the STD values are slightly smaller than in the original calculation. But the relative magnitudes among different regions remain unchanged, and the spatial patterns of variability are still highly consistent with the observational data. Therefore, this adjustment does not affect our main conclusion that the model captures the full variability of spCO₂ well across most regions.

[Figure]

Figure R7 (Figure 8 in main text). Comparison of spCO₂ standard deviation from 1982-2023 between SJTU-AViT and SOCAT. (a) Standard deviation of spCO₂ from the SJTU-AViT reconstruction. (b) Standard deviation of spCO₂ from SOCAT data. (c) Standard deviation ratio, representing the ratio of SJTU-AViT to SOCAT standard deviation (SJTU-AViT divided by SOCAT). (d) Standard deviation bias, showing the difference between the SJTU-AViT and SOCAT standard deviations (SJTU-AViT minus SOCAT). The standard deviation (STD) is quantified as the standard deviation

of residuals after removing long-term trends. In the panels c and d, the SJTU-AViT values are interpolated to match the spatial and temporal locations of SOCAT observations (see detailed computation in section 2.3).

[Figure]

Figure R8 (Figure S5 in supplement section S4). Bias in the standard deviation of $spCO_2$ between SJTU-AViT and SOCAT at each season from 1982 to 2023. (a) MAM (March-May), (b) JJA (June-August), (c) SON (September-November), and (d) DJF (December-February). The standard deviation (STD) is quantified as the standard deviation of residuals after removing long-term trends. The bias is calculated as the difference between SJTU-AViT and SOCAT standard deviations at each season (SJTU-AViT minus SOCAT). Positive values (red) indicate overestimation of variability by SJTU-AViT, while negative values (blue) indicate underestimation. These seasonal biases highlight the model's performance across different seasonal periods and regions. The $spCO_2$ in SJTU-AViT is interpolated to match the SOCAT observation locations and times in the comparison (see detailed computation in section 2.3).

8. L391-396: A presentation issue. The seasonal changes are, physically, attributed to these factors you mentioned. This is based on our understanding of the ocean carbon dynamics rather than being directly learned from ML output. The sentences read like you confirm these dominant factors from your model output. Might consider making it clear that these are not model results. Or, indeed, you could do factor contribution analysis.

We agree that the physical attributions in the original text are based on established oceanographic understanding rather than direct causal inferences from the ML outputs. We have clarified the text in the revised manuscript (now is lines 422-427):

*"Furthermore, the model reasonably reproduces seasonal increases in spCO$_2$ in the North Pacific and North Atlantic (40°-60°N) during Northern Hemisphere winter and early spring. This suggests that the model has likely captured underlying mechanisms, such as the deepening of the winter mixed layer and the entrainment of DIC-rich subsurface waters, which drive seasonal variations in surface ocean pCO$_2$ (Keppler et al., 2020). Conversely, a pronounced seasonal decrease in spCO$_2$ is simulated in the high-latitude Southern Ocean (south of 60°S) during the same period, indicating that the model may also have learned the influence of cooling-driven solubility changes and biological activity on ocean pCO$_2$."*

9. Figure 9: I think what is missing here is to show whether the seasonal phases are consistent compared to SOCAT.

To evaluate whether our reconstruction can accurately capture the seasonal phase observed in SOCAT, we carried out additional analyses comparing the model results with SOCAT climatologies (new supplement section S5.9; see lines 448-453 in the revised manuscript). Specifically:

  (a) Seasonal cycle comparison across ocean basins: We have evaluated the seasonal cycle month-by-month for the global ocean and five major basins, separately for the Northern and Southern Hemispheres. These comparisons demonstrate that the model well reproduces the seasonal cycle of spCO$_2$, with peak and minimum months largely consistent with SOCAT observations (Figs. R9-R10).

  (b) Phase bias evaluation: We produced global maps of the difference in ocean pCO$_2$ peak month and minimum month between SJTU-AViT and SOCAT (in months, range ±6). Across most regions, the phase differences in both peak and minimum months are within ±1 month, with only ~5% of grid points exceeding this threshold (Fig. R11).

  Together, these results indicate that the reconstruction reliably reproduces seasonal phasing. The corresponding text has been added in the revised manuscript section 3.3 (lines 448-453): *"To evaluate the accuracy of the SJTU-AViT in capturing the seasonal phasing of spCO$_2$, we compared it against SOCAT climatology (supplement Figs. S16-S18). Climatological seasonal cycles were evaluated for the global ocean and five major basins, separately for the Northern and Southern Hemispheres. The SJTU-AViT closely reproduces the timing of seasonal maxima and minima in spCO$_2$, generally aligning with SOCAT observations. Global maps of phase differences show that most regions deviate by less than ±1 month, with only ~5% of grid points exceeding this range. These results demonstrate that the reconstruction data reliably captures the observed seasonal phasing."*.

[Figure]

Figure R9 (Figure S16 in supplement section S5). Monthly spCO₂ regional time series for the Northern Hemisphere across different ocean regions from 1982 to 2023. Each panel shows the 12-month mean seasonal cycle for both the model (SJTU-AViT) and SOCAT observations. Peak months are indicated to allow direct comparison of seasonal phasing.

[Figure]

Figure R10 (Figure S17 in supplement section S5). Monthly spCO₂ regional time series for the Southern Hemisphere across different ocean regions from 1982 to 2023. Each panel shows the 12-month mean seasonal cycle for both the model (SJTU-AViT) and SOCAT observations. Peak months are indicated to allow direct comparison of

seasonal phasing.

[Figure]

Figure R11 (Figure S18 in supplement section S5). Grid-scale maps of spCO$_2$ peak- and minimum-month differences (SJTU-AViT − SOCAT, in months, range ±6). For the peak-month difference map, positive values indicate that SJTU-AViT peaks later than SOCAT; for the minimum-month difference map, positive values indicate that SJTU-AViT minimums later than SOCAT. Regions with insufficient observational coverage are masked. These maps provide a spatial assessment of the model's ability to reproduce seasonal maxima and minima timing.

10. Figure 11: Linearly detrended spCO$_2$?
Yes, all spCO$_2$ data shown (now is Fig. 12) have been linearly detrended and deseasonalized. This processing ensures that the composite mean anomalies clearly highlight the typical spCO$_2$ responses associated with El Niño and La Niña events. We have updated the figure caption in the revised manuscript (lines 541-545), now as *"Figure 12. Comparison of spCO$_2$ anomalies during El Niño and La Niña events between SJTU-AViT and multiple data products. Panels (a) and (b) show the composite mean spCO$_2$ anomalies during eight El Niño and seven La Niña events, respectively, as reconstructed by the SJTU-AViT product. Panels (c) and (d) display the corresponding composite mean anomalies from the ensemble mean of eight spCO$_2$ data products. The eight El Niños and seven La Niñas are indicated in the supplement section S2 and S3. The spCO$_2$ anomalies are defined as residuals after removing both long-term trends and seasonal cycles."*.

11. L568-571: PDO-related SST patterns are used in your training; incorporating other indices (e.g., directly using PDO) would be double counting?
Indeed, the PDO signal is already implicitly embedded in the SST fields used as predictors, so directly adding the PDO index could raise concerns about double counting. However, machine learning models are not always efficient at extracting such low-frequency signals, particularly when the observational record is relatively short. In these cases, providing strong or even redundant cues can facilitate the machine learning model representation of decadal variability. In our additional experiments with physical constraints, we found that explicitly highlighting such kind of signals enabled the model to more effectively detect latent signals that are difficult

to capture, thereby improving reconstruction accuracy.

References not in manuscript:

Doney, S. C., Lima, I., Feely, R. A., Glover, D. M., Lindsay, K., Mahowald, N., Moore, J. K., Wanninkhof, R.:  Mechanisms governing interannual variability in upper-ocean inorganic carbon system and air–sea $CO_2$ fluxes: Physical climate and atmospheric dust. Deep-Sea Res. Pt. II, 56, 640655. https://doi.org/10.1016/j.dsr2.2008.12.006, 2009.

Liao, E., Resplandy, L., Liu, J., and Bowman, K. W.: Future Weakening of the ENSO Ocean Carbon Buffer Under Anthropogenic Forcing, Geophys. Res. Lett., 48, e2021GL094021, https://doi.org/10.1029/2021GL094021, 2021.

Stock, C. A., Dunne, J. P., Fan, S., Ginoux, P., John, J., Krasting, J. P., Laufkötter, C., Paulot, F., and Zadeh, N.: Ocean Biogeochemistry in GFDL's Earth System Model 4.1 and Its Response to Increasing Atmospheric $CO_2$, J. Adv. Model. Earth Syst., 12, e2019MS002043, https://doi.org/10.1029/2019MS002043, 2020.